



# Models for polythermal ice sheets and glaciers

Ian J Hewitt[1] and Christian Schoof[2]

[1]University of Oxford, Oxford, UK
[2]University of British Columbia, BC, Canada

*Correspondence to:* Ian Hewitt (hewitt@maths.ox.ac.uk)

**Abstract.** Polythermal ice sheets and glaciers contain both cold ice and temperate ice. We present two new models to describe the temperature and water-content of such ice masses, accounting for the possibility of gravity- and pressure-driven water drainage according to Darcy's law. Both models are based on the principle of energy conservation; one additionally invokes the theory of viscous compaction to calculate pore water pressure, and the other involves a modification of existing enthalpy gradient methods to include gravity-driven drainage. The models consistently predict the evolution of temperature in cold ice, and of water content in temperate ice. Numerical solutions are described, and a number of illustrative test problems are presented, allowing comparison with existing methods. The suggested models are simple enough to be incorporated in existing ice-sheet models with little modification.

## 1 Introduction

This paper concerns the modelling of temperature and water content in polythermal ice masses. These are ice sheets and glaciers that contain both cold ice, below its pressure melting temperature, and temperate ice, which is a two-phase mixture of ice and water at the melting temperature. Such models are an important component of ice-sheet simulations because the rheology of ice, and hence its ability to flow, depend strongly on temperature and water content. The specific focus of this paper is on models that allow for gravity-driven drainage of meltwater in temperate ice.

There is already an extensive literature on the subject of polythermal ice (Fowler and Larson, 1978; Hutter, 1982; Fowler, 1984; Greve, 1997; Aschwanden et al., 2012). Ice temperature is governed by an advection-diffusion equation, with dissipation due to viscous creep acting as a heat source. If the temperature reaches the melting point, further dissipative heating leads to internal melting, and the resulting temperate ice can be described using mixture theory. Combining both options leads to a free-boundary problem for temperature and water content (the two are often combined, as enthalpy).

A key ingredient of this theory is a constitutive law for water transport in the temperate ice. There is little experimental constraint on this, but it is thought to be modelled either by Darcy's law or by a diffusive flux, or perhaps a combination. Existing computational models adopt the diffusive approach, and in this paper we explore the alternative that uses Darcy's law. This has the distinct





advantage of allowing gravity-driven drainage, due to the ice-water density difference. It has the disadvantage of requiring assumptions to be made about temperate ice permeability (we take an

increasing power of the porosity), and about pore water pressure.

In a companion paper (Schoof and Hewitt, 2016), we presented the derivation and simplification of a 'compaction pressure' model, and explored mathematical properties of its solutions in detail. In this paper we describe the approximate version of this model that we consider suitable for numerical computations. We compare it to a standard 'enthalpy gradient' model in which liquid water transport

is proportional to the gradient of porosity, and to a modified version of the enthalpy gradient model that also includes the gravitational component of Darcy's law (obtained by assuming pore pressure is equal to ice pressure).

Enthalpy gradient models have become widely used in the literature (eg. Aschwanden et al., 2012; Brinkerhoff and Johnson, 2013; Seroussi et al., 2013; Wilson and Flowers, 2013; Kleiner et al., 2015;

Blatter and Greve, 2015). Both our compaction pressure model and our modified enthalpy gradient model provide alternatives that include gravity-driven water transport. This difference is important when considering the energy distribution in the ice sheet, because meltwater can transport significant amounts of latent heat.

In section 2 we lay out the model equations and the alternative parameterisations of water trans-

port. Section 3 describes the algorithm that has been used to solve the models, and in section 4 we consider three example problems, for which we apply the different models and compare their behaviour. We conclude with a discussion and conclusions in section 5.

## 2   Model equations

### 2.1   Ice flow

We assume that ice velocity satisfies the usual Stokes equations,

$$\nabla \cdot \mathbf{u} = 0, \tag{1}$$

$$\frac{\partial \tau_{ij}}{\partial x_j} - \frac{\partial p}{\partial x_i} = -\rho g_i, \tag{2}$$

with a power-law rheology described by

$$\tau_{ij} = A^{-1/n} \dot{\varepsilon}^{1/n-1} \dot{\varepsilon}_{ij}, \qquad \dot{\varepsilon}_{ij} = \frac{1}{2} \left( \frac{\partial u_i}{\partial x_j} + \frac{\partial u_j}{\partial x_i} \right), \qquad \dot{\varepsilon} = \sqrt{\dot{\varepsilon}_{ij} \dot{\varepsilon}_{ij}/2}, \tag{3}$$

Here $\mathbf{u} = u_i$ is the ice velocity, $\tau_{ij}$ is the deviatoric stress tensor, $p$ is the ice pressure, $\rho$ is the ice density, $\mathbf{g} = g_i$ is the gravitational acceleration, and $\dot{\varepsilon}_{ij}$ the strain rate tensor, with second invariant $\dot{\varepsilon}$. The summation convention is used for repeated indices. The rheological parameters $A$ and $n$ potentially depend on temperature $T$ and porosity (water content) $\phi$.



It is a basic assumption of this study that these equations provide an adequate description of ice deformation even when the temperate 'ice' is actually a two-phase mixture of ice and pore water. As discussed by Aschwanden et al. (2012), this is not strictly true, since drainage of water from the ice renders the mixture slightly compressible. Ignoring the ice-water density difference (which we do throughout, except when multiplied by gravity), the expression for mass conservation (1) should
become

$$\nabla \cdot \mathbf{u} = -\nabla \cdot \mathbf{j}, \tag{4}$$

where $\mathbf{j} = \phi(\mathbf{u}_w - \mathbf{u})$ is the relative water flux, discussed further below (here $\mathbf{u}_w$ is the water velocity). There are corresponding alterations to the stress balance (Schoof and Hewitt, 2016). Provided internal melting is not too large, however, the water flux is expected to be small compared to the
ice flux so that the right hand side of (4) may be reasonably ignored. We therefore suppose that ice deformation is governed everywhere by (1)-(3). A primary reason for making this approximation is to avoid the thermodynamic model interfering with the basic Stokes-flow solution, which forms the backbone of most ice-sheet models; however, the approximation can also be reasoned with systematic scaling arguments (Schoof and Hewitt, 2016). Alternatively, one can maintain the full expression
(4), and we comment on this further in section 5.

## 2.2   Energy

Conservation of energy is expressed as

$$\rho c \left( \frac{\partial T}{\partial t} + \mathbf{u} \cdot \nabla T \right) = \nabla \cdot (k \nabla T) + \tau_{ij} \dot{\varepsilon}_{ij} - \rho_w L m, \tag{5}$$

where $c$ is the specific heat capacity, $k$ is the thermal conductivity, $\rho_w$ is the water density, $L$ is the
latent heat, and $m$ is the internal melting rate. If the temperature $T$ is below the pressure melting point $T_m$ then $m$ is zero and this is the standard heat equation. If the temperature reaches $T_m$ (it is not allowed to exceed this value) this equation determines the melting rate $m$, which provides a source term to the moisture equation,

$$\frac{\partial \phi}{\partial t} + \mathbf{u} \cdot \nabla \phi + \nabla \cdot \mathbf{j} = m. \tag{6}$$

Here $\mathbf{j}$ is the water flux, for which three different models are discussed below. In writing (6) we have again made use of the approximation that the ice is incompressible (Schoof and Hewitt, 2016).

   Following standard practice, it is convenient to combine (5) and (6) as a single equation that holds in both cold and temperate ice. We define the enthalpy in this context as

$$h = \rho c(T - T_{ref}) + \rho_w L \phi, \tag{7}$$





where $T_{ref}$ is a fixed reference temperature ($T_{ref} = 273$K), and the inverse relationship for temperature and porosity as a function of enthalpy is given by

$$T = \begin{cases} T_{ref} + h/\rho c \\ T_m \end{cases} \qquad \phi = \begin{cases} 0 & h < \rho c(T_m - T_{ref}) \\ [h - \rho c(T_m - T_{ref})]/\rho_w L & h \geq \rho c(T_m - T_{ref}) \end{cases}. \qquad (8)$$

The energy and moisture equations (5) and (6) are then combined as

$$\frac{\partial h}{\partial t} + \mathbf{u} \cdot \nabla h + \nabla \cdot \mathbf{Q} = \tau_{ij} \dot{\varepsilon}_{ij}, \qquad \mathbf{Q} = \begin{cases} -k\nabla T & h < \rho c(T_m - T_{ref}) \\ -k\nabla T_m + \rho_w L \mathbf{j} & h \geq \rho c(T_m - T_{ref}) \end{cases}. \qquad (9)$$

Enthalpy is transported by advection and conduction in the cold regions, and by advection and water transport in the temperate regions (as well as conduction due to variations of the pressure-dependent melting point).

The jump condition (Stefan condition) that naturally arises from (9) at cold-temperate interfaces is

$$\rho_w L \phi (\mathbf{u} - \mathbf{v}) \cdot \mathbf{n} = -k\nabla T^- \cdot \mathbf{n} + k\nabla T_m \cdot \mathbf{n} - \rho_w L \mathbf{j} \cdot \mathbf{n}, \qquad (10)$$

where $\mathbf{v}$ is the velocity of the interface, $\mathbf{n}$ is the normal to the interface, and $^-$ refers to the cold side. The left hand side here represents the latent heat released as water advected through the interface freezes, which, if non-zero, gives rise to a discontinuity in the temperature gradients on the right hand side. The mass conservation equation (4) gives rise to a corresponding jump condition,

$$\mathbf{j} \cdot \mathbf{n} = 0, \qquad (11)$$

(it is assumed that the ice velocity $\mathbf{u}$ must be continuous across the interface, else there would be a stress singularity). In our compaction pressure model we make use of (11), and the last term of (10) is therefore zero. In the enthalpy gradient models, we are forced to ignore (11), which must be justified on the basis that we have already approximated the mass conservation equation by (1). The
presence of the last term in (10) then leads to a (typically small) adjustment in the location of the cold-temperate interface in this case (see section 4.1). Note that the interface condition (10) is not used explicitly in our approach; but the condition is implicit in the conservative solution of (9).

### 2.3 Water transport

In the following subsections, we describe three specific models for the water flux $\mathbf{j}$. Together with (8)
and (9), these provide a complete thermal model from which to compute temperature and porosity. As noted above $\mathbf{j} = \phi(\mathbf{u}_w - \mathbf{u})$, where $\mathbf{u}_w$ is the water velocity, and prescribing a law for $\mathbf{j}$ is equivalent to prescribing one for $\mathbf{u}_w$; we choose to work with the flux, from which the velocity could be inferred.



### 2.3.1 Compaction pressure model

This model assumes that water transport occurs according to Darcy's law, driven by gravity and pressure gradients. We write

$$\mathbf{j} = \frac{k_0 \phi^\alpha}{\eta_w} \left( \rho_w \mathbf{g} - \nabla p_w \right), \tag{12}$$

where $k_0$ is a permeability factor, $\eta_w$ is the viscosity of the water, and $p_w$ is the pore pressure. There is uncertainty over an appropriate value for $k_0$ (which likely depends on grain size and impurity

content); the exponent $\alpha$ is likely to be between 2 and 3 (Nye and Frank, 1973). The pore water pressure $p_w$ is typically reduced from the ice pressure $p$, and the difference is referred to as the effective pressure or compaction pressure, $p_e \equiv p - p_w$. Crucially, this pressure difference is related to the ice compaction rate, by

$$p_e = -\zeta \nabla \cdot \mathbf{u} = \zeta \nabla \cdot \mathbf{j}, \tag{13}$$

where $\zeta = \eta/\phi$ is the bulk ice viscosity, given in terms of the effective viscosity $\eta = \frac{1}{2} A^{-1/n} \dot{\varepsilon}^{1/n-1}$ and the porosity (Fowler, 1984); the second equality follows from (4). (Note that in calculating $p_e$ here we care about how small the compaction rate is, so we must relate it to the divergence $\nabla \cdot \mathbf{j}$ despite approximating the ice as incompressible in (1).) The expression in (13) is related to the expression derived by Nye (1953) for the rate of closure of a borehole due to confining pressure.

If we approximate the ice pressure as hydrostatic, $\nabla p \approx \rho \mathbf{g}$ (the deviatoric corrections from (2) are assumed to be small, though it would also be straightforward to include them), we can rewrite (12) as

$$\mathbf{j} = \frac{k_0 \phi^\alpha}{\eta_w} \left( (\rho_w - \rho) \mathbf{g} + \nabla p_e \right). \tag{14}$$

Substituting this into (13) then yields

$$\phi p_e = \frac{\eta k_0}{\eta_w} \nabla \cdot \left[ \phi^\alpha \left( (\rho_w - \rho) \mathbf{g} + \nabla p_e \right) \right]. \tag{15}$$

The combination of (14) and (15) provide the necessary ingredients to determine $\mathbf{j}$ in (9). Compared to the models in section 2.3.2 and 2.3.3, there is an additional variable, $p_e$, to be solved for only in the temperate region. For a given porosity field, (15) is a linear elliptic equation to determine this pressure. The appropriate boundary conditions are $\mathbf{j} \cdot \mathbf{n} = 0$ on cold-temperate interfaces, and either

prescribed pressure or prescribed flux on exterior boundaries. Typically we set $\mathbf{j} \cdot \mathbf{n} = 0$ (or some other value) at the glacier surface, and $p_e = N + \tau_{nn}$ at the glacier bed, where $N$ is the effective pressure in the subglacial drainage system and $\tau_{nn}$ is the deviatoric normal stress on the bed. More extensive discussion of boundary conditions is given by Schoof and Hewitt (2016).

### 2.3.2 'Standard' enthalpy gradient model

A common approach is to suppose that the water transport is diffusive, so

$$\mathbf{j} = -\nu \nabla \phi, \tag{16}$$





for some coefficient $\nu$. In this case (9) is simply an advection-diffusion equation with variable diffusivity depending on the value of $h$. There is little knowledge about how large the coefficient $\nu$ should be. The choice $\nu = k/\rho c$ would yield the same diffusivity for enthalpy in both cold and temperate

domains, and $\nu$ is typically taken to be rather smaller than this. Indeed, with a small value of $\nu$ this term may be treated as a regularization of the problem with $\nu = 0$, in which water is simply advected with the ice. In that case, the energy equation (9) is a hyperbolic equation on the temperate regions, meaning that the porosity (and hence enthalpy) need not be continuous at the cold-temperate interfaces. The diffusive term prevents such discontinuities (which might be desirable for numerical

reasons).

This model does not allow for significant drainage, and can result in the build up of large amounts of water in the ice. A cap on porosity, and removal of additional water, are often implemented, by adding a term $-\rho_w L D(\phi)$ to the right hand side of (9), where $D(\phi)$ is a 'drainage function' with units $s^{-1}$ (eg. Greve, 1997; Aschwanden et al., 2012).

### 165 2.3.3 'Modified' enthalpy gradient model

Motivated by the desire to capture gravity-driven drainage of the meltwater, this model takes the gravitational component of Darcy's law from (14), adding to this a diffusive flux as in (16). We write

$$\mathbf{j} = -\nu \nabla \phi + \frac{k_0 \phi^\alpha (\rho_w - \rho) \mathbf{g}}{\eta_w}. \tag{17}$$

This model has the advantage of allowing water to drain from the ice, as in the compaction pressure model, but without the need to solve an additional equation for $p_e$. As demonstrated below, the models often yield similar behaviour except in boundary layers. Both models are distinct from the drainage function $D(\phi)$ mentioned above, because they explicitly model the water transport through the ice. The model in (17) can be derived from (14) if it is assumed that the compaction pressure

depends upon porosity, $p_e(\phi)$, in which case $\nu = -k_0 \phi^\alpha p_e'(\phi)/\eta_w$. Such a relationship between pressure and $\phi$ may be justified on the basis of surface energy effects, particularly if the grain size is small (Bercovici et al., 2001; Schoof and Hewitt, 2016).

## 3 Numerical method

This section explains the numerical method used to solve the energy model consisting of (8) and

(9) together with either (14) and (15), (16), or (17). The problem is coupled to the solution for ice velocity from (1)-(3). Alternatively, those full Stokes equations can be replaced by standard approximations (see, eg. Schoof and Hewitt, 2013). However, for the test problems considered in this paper we will not concern ourselves with the coupled ice-flow problem and simply take the ice velocity as given.





We employ a first-order finite volume discretisation with a simple operator splitting method. Temperature, porosity, and effective pressure are discretized on a regular grid, while ice velocity and water flux are defined on a staggered grid. Using a harmonic average to define the permeability on the staggered grid means that the condition $\mathbf{j} \cdot \mathbf{n} = 0$ is naturally applied at any cold-temperate boundary.

At each timestep, (9) is advanced for $h$ with a combination of implicit and explicit discretization. Specifically, the domain is partitioned according to the current value of $h$ into temperate and cold regions, and the diffusion operators for temperature/porosity are discretised implicitly on the cold/temperate domain, respectively, according to this partition. All other terms in $\nabla \cdot \mathbf{Q}$ are discretised explicitly. Having advanced $h$, the new values of $T$ and $\phi$ are defined by (8) and, in the compaction pressure model, the pressure is calculated from the updated $\phi$ by solving (15). It is possible that a more accurate time-stepping procedure could be implemented, but this method has the advantage of simplicity.

For clarity, we summarise the time discretization here, with superscripts referring to the timestep (ignore the hats on some of the variables, which are included for later reference):

$$\frac{h^{(n)} - h^{(n-1)}}{t^{(n)} - t^{(n-1)}} + \mathbf{u} \cdot \nabla h^{(n)} - \hat{\nabla} \cdot (k \hat{\nabla} T) + \rho_w L \nabla \cdot \hat{\mathbf{j}} = \tau_{ij} \dot{\varepsilon}_{ij}, \tag{18}$$

where

$$T = \begin{cases} h^{(n)}/\rho c & h^{(n-1)} < \rho c (T_m - T_{ref}) \\ T_m & h^{(n-1)} \geq \rho c (T_m - T_{ref}) \end{cases}, \tag{19}$$

and, for the compaction pressure model,

$$\hat{\mathbf{j}} = \frac{k_0 \phi^{(n-1)\alpha}}{\eta_w} \left( (\rho_w - \rho) \hat{\mathbf{g}} + \hat{\nabla} p_e^{(n-1)} \right), \quad \phi^{(n-1)} p_e^{(n-1)} = \frac{\eta k_0}{\eta_w} \hat{\nabla} \cdot \left[ \phi^{(n-1)\alpha} \left( (\rho_w - \rho) \hat{\mathbf{g}} + \hat{\nabla} p_e^{(n-1)} \right) \right], \tag{20}$$

while, for the modified enthalpy gradient method,

$$\hat{\mathbf{j}} = -\nu \hat{\nabla} \phi + \frac{k_0 \phi^{(n-1)\alpha} (\rho_w - \rho) \hat{\mathbf{g}}}{\eta_w}, \qquad \phi = \begin{cases} 0 & h^{(n-1)} < \rho c (T_m - T_{ref}) \\ h^{(n)}/\rho_w L & h^{(n-1)} \geq \rho c (T_m - T_{ref}) \end{cases}. \tag{21}$$

The standard enthalpy method is identical except that the gravitational term in (21) is ignored. Drainage is included in the standard enthalpy method by adding $-\rho_w L D(\phi^{(n-1)})$ to the right hand side of (18), with the function $D(\phi)$ taken from Figure 4 of Aschwanden et al. (2012); $D(\phi)$ is zero for $\phi < 1\%$, and then increases in a piecewise linear fashion.

Boundary conditions are discussed by Schoof and Hewitt (2016) and also by Aschwanden et al. (2012), and we avoid going into the details here. Briefly, for the compaction pressure model or if $\nu = 0$, the enthalpy equation (9) is elliptic on cold domains and hyperbolic on temperate domains, so





**Table 1.** Values of parameters used for example solutions. The value of $\nu$ is 100 times smaller than $k/\rho c$.

| | | | |
|---|---|---|---|
| $\rho$ | $916 \text{ kg m}^{-3}$ | $n$ | $3$ |
| $c$ | $2009 \text{ J kg}^{-1} \text{ K}^{-1}$ | $A$ | $2.4 \times 10^{-24} \text{ Pa}^{-3} \text{ s}^{-1}$ |
| $k$ | $2.1 \text{ W m}^{-1} \text{ K}^{-1}$ | $\alpha$ | $2$ |
| $\rho_w$ | $1000 \text{ kg m}^{-3}$ | $k_0$ | $10^{-12} \text{ m}^2$ |
| $L$ | $3.34 \times 10^5 \text{ J kg}^{-1}$ | $\eta_w$ | $1.8 \times 10^{-3} \text{ Pa s}$ |
| $g$ | $9.8 \text{ m}^2 \text{ s}^{-1}$ | $\nu$ | $1.1 \times 10^{-8} \text{ m}^2 \text{ s}^{-1}$ |

requires conditions on temperature (typically Robin-type conditions) at cold boundaries, and condi-
tions on porosity at inflowing temperate boundaries. If $\nu > 0$, an additional condition is required on
porosity at outflowing temperate boundaries (where it gives rise to a boundary layer).

     This algorithm has been tested against steady-state solutions calculated using a shooting method,
and asymptotic methods (Schoof and Hewitt, 2016). In all of the solutions shown here the ice velocity
is prescribed; solutions with fully coupled ice velocity (dependent on temperature and water content)
have not yet been tested, but we see no reason why the same method should not work. Coupling
occurs through the advection term $\mathbf{u}$ and the dissipation term $\tau_{ij} \dot{\varepsilon}_{ij}$ in (18), and through the ice
viscosity $\eta$ in (20).

     Since ice sheets and glaciers typically have a shallow aspect ratio, it is common to approximate
conduction terms using the fact that vertical ($z$) derivatives are much larger than horizontal ($x$ and
$y$) derivatives. For the same reason we anticipate that the dominant water transport (besides advec-
tion) occurs in the vertical direction. We therefore approximate many of the divergence and gradient
operators in the model by taking only their vertical components; specifically, we take only the $z$
components of the terms adorned by hats in (18)-(21).

     In section 4, we consider domains of non-uniform ice thickness, for which the numerical cal-
culation uses a spatially-variable stretch of the vertical coordinate. This standard coordinate trans-
formation introduces additional advection terms into the equations as well as rescaling the vertical
derivatives.

     The model can accommodate a pressure dependent melting point $T_m = T_{ref} - \gamma(p - p_{ref})$, where
$\gamma \approx 7.5 \times 10^{-8} \text{ K Pa}^{-1}$ is the Clapeyron slope, but the solutions shown here all have $T_m \equiv 0$ C,
which allows for easier visualisation of the results. Including pressure dependence leads to slightly
enhanced internal melting when ice is advected towards deeper ice, and reduced melting (or re-
freezing) when advection is towards shallower ice. Strictly, the melting point should depend on
pore pressure $p_w$ rather than ice pressure, and preliminary investigations suggest that the resulting
interactions can lead to some unexpected behaviour, but this is beyond our current scope.



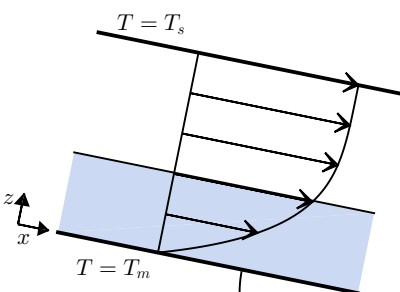

**Figure 1.** Parallel-sided slab setup. Arrows indicate the bed-parallel velocity, and shading indicates the region of temperate ice (to be found).

## 4 Example solutions

### 4.1 Parallel-sided slab

A simple one-dimensional test problem for these models is the case of a parallel-sided slab of ice, as seen in Fig 1. Aligning coordinates parallel and perpendicular to the slope, the ice velocity is $\mathbf{u} = (u, w)$, with

$$u = \frac{2A(\rho g S)^n}{n+1}\left(H^{n+1} - (H-z)^{n+1}\right), \tag{22}$$

where $H$ is the ice thickness, $S = \sin\theta$ is the slope, and $A$ is treated as constant (Greve and Blatter, 2009; Kleiner et al., 2015). The corresponding effective viscosity and viscous dissipation rate for this problem are

$$\eta = \frac{1}{2A(\rho g S)^{n-1}(H-z)^{n-1}}, \qquad \tau_{ij}\dot{\varepsilon}_{ij} = 2A(\rho g S)^{n+1}(H-z)^{n+1}. \tag{23}$$

For a genuinely parallel moving slab the normal velocity $w$ is zero; however we also consider cases where $w$ is non-zero (but uniform), corresponding to the ice being accumulated at the top surface and melted from the bottom surface (if $w$ is negative), or vice versa (if $w$ is positive). We take $\mathbf{g} = -g\mathbf{e}_z$, ignoring the slight correction due to the slope.

The top surface $z = H$ is prescribed to be at temperature $T = T_s < T_m$, and the bottom surface $z = 0$ is prescribed to be at the melting point $T = T_m$. If viscous dissipation is large enough, this setup gives rise to a layer of temperate ice at the bottom of the slab. In the case of a positive normal velocity $w$, the additional condition $\phi = \phi_m$ is prescribed there (the porosity of frozen-on ice, although it may be better to treat this as a numerical test problem rather than to ascribe too much physical meaning to it). For the enthalpy gradient methods, this condition on $\phi$ is needed regardless of the direction of ice flow.





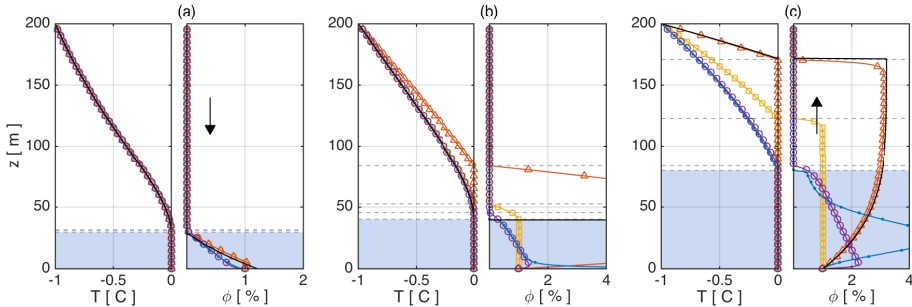

**Figure 2.** Steady-state solutions for the parallel-sided slab, showing temperature and porosity as a function of height. Normal velocities are (a) $w = -0.2$ m y$^{-1}$, (b) $w = 0$ m y$^{-1}$, and (c) $w = 0.2$ m y$^{-1}$. In each case, orange triangles show the standard enthalpy gradient method, yellow squares show the standard enthalpy gradient model method with drainage, purple circles show the modified enthalpy method, and blue dots show the compaction pressure model. Solid black lines show the semi-analytical solutions for the case of no relative water transport. Shading shows the region of temperate ice for the compaction pressure model. Parameter values are as in Table 1, together with $H = 200$ m, $S = \sin 4°$, $T_s = -1°C$, $T_m = 0°C$, $\phi_m = 1\%$.

One of the reasons for exploring this simple setup is that semi-analytical solutions are possible, allowing the algorithms to be tested. Constructing these solutions involves straightforward algebraic manipulations and solution of a transcendental equation for the position of the cold-temperate transition (details may be found, for example, in Greve and Blatter (2009)).

Figure 2 shows the comparison between the three models: compaction pressure, standard enthalpy gradient, and modified enthalpy gradient. The numerical solutions are all run to the steady state, and we compare them with the semi-analytical solution for no relative water transport ($\mathbf{j} = 0$). This solution provides a clear illustration of the different behaviour that can occur at the cold-temperate interface as a result of (10): for downward moving ice (referred to as a melting interface by Greve

(1997)), the temperature gradient is zero on the cold side and the porosity increases continuously from zero on the temperate side (panel (a)); for upward moving ice (a freezing interface), the porosity is finite on the temperate side and there is a corresponding finite temperature gradient on the cold side (panel (c)); and for neither upward nor downward advection, the temperature gradient is zero but there can be finite porosity on the temperate side (in fact, there is no steady solution for the

no-water-flux problem in this case).

The standard enthalpy gradient model closely approximates the no-water-flux solution for both downward and upward moving ice, the diffusive term naturally smoothing out the discontinuity in porosity at the freezing interface (provided $\nu$ is small enough). In the case $w = 0$ (panel (b)), it produces a quite different result.

For the case of downwards moving ice (panel (a)), the location of the cold-temperate boundary is almost identical between all the models. This is reassuring, since the location of the interface in





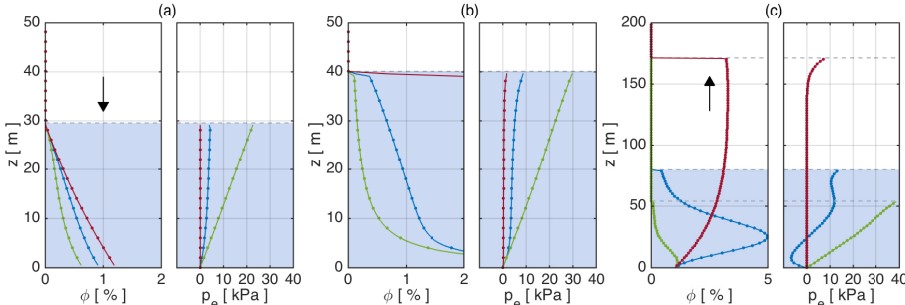

**Figure 3.** Steady-state solutions for the parallel-sided slab with the compaction pressure model, showing porosity and effective pressure as a function of height. Normal velocities are (a) $w = -0.2$ m y$^{-1}$, (b) $w = 0$ m y$^{-1}$, and (c) $w = 0.2$ m y$^{-1}$. In each case blue dots show the same solution as in Figure 2, purple dots are for permeability 100 times smaller (the porosity then approaches the no-water-flux solution), and green dots are for permeability 100 times larger. For large permeability, the pore pressure is essentially hydrostatic, so the effective pressure is linear, $\partial p_e / \partial z \approx (\rho_w - \rho) g$ from (14).

this case should be determined by the twin conditions $T = T_m$, $\nabla T \cdot \mathbf{n} = 0$, and is thus independent of what happens in the temperate region (Schoof and Hewitt, 2016). The effect of gravity-driven drainage is to reduce the porosity compared to the no-water-flux solution, and this reduction is es-

sentially the same in both the compaction pressure and the modified enthalpy gradient models, except very close to the bottom.

For the case $w = 0$ (panel (b)), the location of the cold-temperate interface in the compaction pressure and modified enthalpy gradient models is close to that for the no-water-flux solution (still determined by $T = T_m$, $\nabla T \cdot \mathbf{n} = 0$). The porosity is again very similar between these two models,

except in a boundary layer at the bottom where the condition on effective pressure or on porosity must be satisfied. The solution is of course completely different from the standard enthalpy gradient model, which has much larger porosities in this case (in the absence of any drainage).

For the case of upwards moving ice (panel (c)), the location of the cold-temperate boundary in the two models with gravity-driven drainage is now different from the no-water-flux solution. This

is because the drainage results in lower porosities, and thus less advected latent heat transport than in the no-water-flux case; the jump condition (10) therefore requires a smaller temperature gradient on the cold side of the interface. The location of the interface is again very similar between the compaction pressure and modified enthalpy gradient models. The porosity distribution is less similar in this case, on which we comment further in section 5.

The effective pressure in the compaction pressure model is shown in Figure 3, where we also demonstrate the effect of changing the permeability constant $k_0$. For small permeability the solution approaches the no-water-flux solution as expected; the effective pressure in this case is mostly small, meaning that the pore pressure is essentially the same as the ice pressure. For large permeability



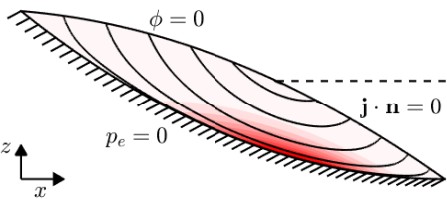

**Figure 4.** Temperate glacier setup, showing boundary conditions for the compaction pressure model. Solid lines show ice streamlines and red shading shows the magnitude of viscous dissipation. Here $Z_b(x) = 500(1 - x/L) - 125(1 - x/L)(x/L)$ m and $Z_s(x) = 500(1 - x/L) + 100(1 - x/L)(x/L)$ m, with $L = 5$ km.

constant the predicted porosities are small, because the water easily drains from the ice. The effective

pressure varies linearly with depth, indicating that the pore pressure is essentially hydrostatic (rather than cryostatic); in this case the drainage rate is actually not controlled by the buoyancy term $(\rho_w - \rho)\mathbf{g}$, but rather by the rate of viscous compaction due to the effective pressure. The modified enthalpy gradient model is not a good approximation of the compaction pressure model in this case.

Overall, we see that inclusion of gravity-driven drainage with either model reduces the porosity,

as expected, and alters the location of the cold-temperate interfaces due to changes in the advected latent heat. These basic features of the models carry over into more complex problems in more dimensions.

### 4.2  Temperate glacier

We illustrate the effect of water transport further by considering an idealised mountain glacier as seen

in Figure 4, with bed elevation $z = Z_b(x)$ and surface elevation $z = Z_s(x)$. Ice velocity $\mathbf{u} = (u, w)$ is taken to follow the non-sliding shallow ice approximation, so $u$ is given by (22) with $S = -\partial Z_s/\partial x$ and $H = Z_s - Z_b$, and $w$ follows from mass conservation with zero vertical velocity at the bed,

$$w = -\int_0^z \frac{\partial u}{\partial x}\, \mathrm{d}z. \tag{24}$$

The effective viscosity and dissipation rate are given by (23) (because of the shallow aspect ratio,

only the contributions of vertical shear, $\partial u/\partial z$, are included; other components are smaller).

Both the upper surface $z = Z_s$ and the lower surface $z = Z_b$ are prescribed to be at the melting temperature $T = T_m$, so the entire glacier is temperate. The conditions imposed on water content differ according to the model. For the enthalpy gradient models we impose $\phi = 0$ on both upper and lower surfaces. For the compaction pressure model, $\phi = 0$ is prescribed on the inflowing part of the

upper surface (the accumulation region), and $\mathbf{j} \cdot \mathbf{n} = 0$ is imposed on the outflowing part (the ablation region). On the lower surface, the effective pressure $p_e$ is prescribed as a fixed value, which we set to 0 for illustration.





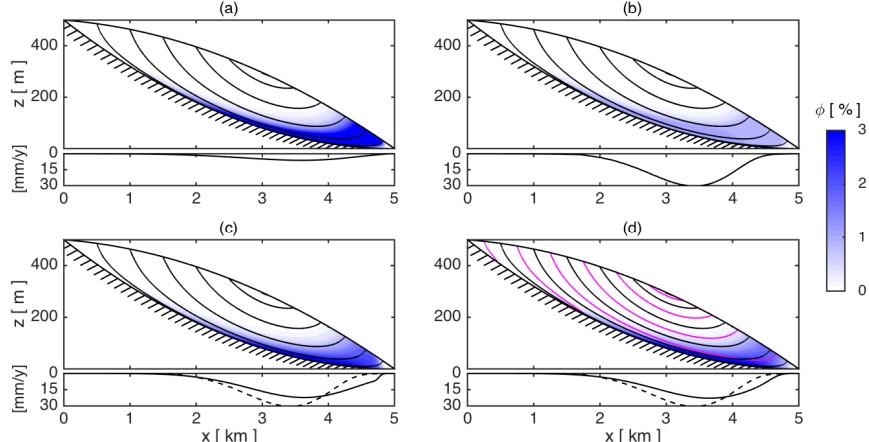

**Figure 5.** Temperate glacier solutions for (a,b) the standard enthalpy gradient model without and with drainage, (c) modified enthalpy gradient model, and (d) compaction pressure model. Shading shows porosity. Solid black lines show ice streamlines. Lower panels show water flux from the ice at the bed (in (b), this includes the vertically-integrated drainage $\int D(\phi)\,\mathrm{d}z$); dashed lines are replicas of that from (b) for comparison. Pink lines in (d) show liquid water streamlines. Parameter values are as in Table 1.

It is worth pointing out that in a more complete model, an energy balance and firn compaction model at the upper surface would prescribe both $\phi$ (in the accumulation region) and water flux $\mathbf{j} \cdot \mathbf{n}$.

At the lower surface, the effective pressure would be determined by conditions in the subglacial drainage system.

Figure 5 shows the steady-state solutions for the standard enthalpy gradient model, with and without a drainage function, the modified enthalpy gradient model, and the compaction pressure model. The lower panels in each case show the water flux from the base of the ice (into the un-modelled

subglacial drainage system).

We see that the compaction pressure model (panel (d)) and the modified enthalpy gradient model (panel (c)) give very similar results. The meltwater flux at the bed is of course larger than for the standard enthalpy gradient model. The flux is similar in the two gravity-driven drainage models to what reaches the bed if the drainage function is included in the standard model (panel (b)), but it

does have slight differences; notably, it is transported further down glacier, because the water is being advected as well as draining vertically, rather than draining instantaneously to the bed. Of course, the actual amount that drains is sensitive to the drainage function $D(\phi)$, or the permeability $k_0 \phi^\alpha$.





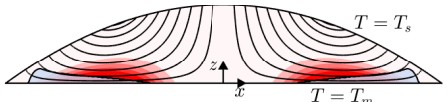

**Figure 6.** Ice cap setup. Solid lines show ice streamlines, red shading shows the magnitude of viscous dissipation, and blue shading shows the region of temperate ice (to be found). Here $Z_b(x) = 0$ m and $Z_s(x) = 1500(1 - (x/L)^2)$ m, with $L = 100$ km.

### 4.3 Ice cap

For the final example we take a symmetric ice-sheet profile as seen in Figure 6, with surface elevation $z = Z_s(x)$. Ice velocity $\mathbf{u} = (u, w)$ is again taken to follow the non-sliding shallow ice approximation, so $u$ is given by (22) with $S = -\partial Z_s/\partial x$, and $w$ follows from (24). The effective viscosity and dissipation rate are given by (23).

The upper surface $z = Z_s$ is prescribed to be at temperature $T = T_s < T_m$, and the lower surface
$z = 0$ is at the melting point $T = T_m$. For the enthalpy gradient models, we also prescribe $\phi = 0$ on the lower surface. Since the ice velocity there is zero, no condition on the porosity is needed for the compaction pressure model, and we just have to prescribe the effective pressure $p_e$, which is set to 0. As in section 4.2, a more complete model could be coupled with a surface firn model and a subglacial drainage model. One could also compute the temperature below the bed, and thus have regions of
the bed that are frozen (if the geothermal heat flux is large enough the entire bed will be temperate, as assumed here, though the ice above is not necessarily temperate, as seen in the solutions).

Figure 7 shows the steady-state solutions for the standard enthalpy gradient model, with and without a drainage function, the modified enthalpy gradient model, and the compaction pressure model. Figure 8 shows similar solutions with a much warmer surface; probably a rather unrealistic scenario,
but it helps to illustrate the difference between the models.

The predicted region of temperate ice is similar between the models, since it is largely controlled by the location of dissipative heating. However, there are some differences: notably the downstream end of the temperate region extends closer to the ice surface in the standard enthalpy gradient method without drainage, because of the large advected latent heat transport, which is all released at the
freezing interface. This is essentially the same behaviour as seen in the upward moving parallel slab in Figure 2(c).

The distribution of porosity is very similar between the modified enthalpy gradient method and the compaction pressure model. It increases towards the bed because of the accumulated drainage of water from above, whereas the porosity distribution in the standard model with drainage is almost
uniform because it is evacuated directly to the bed.





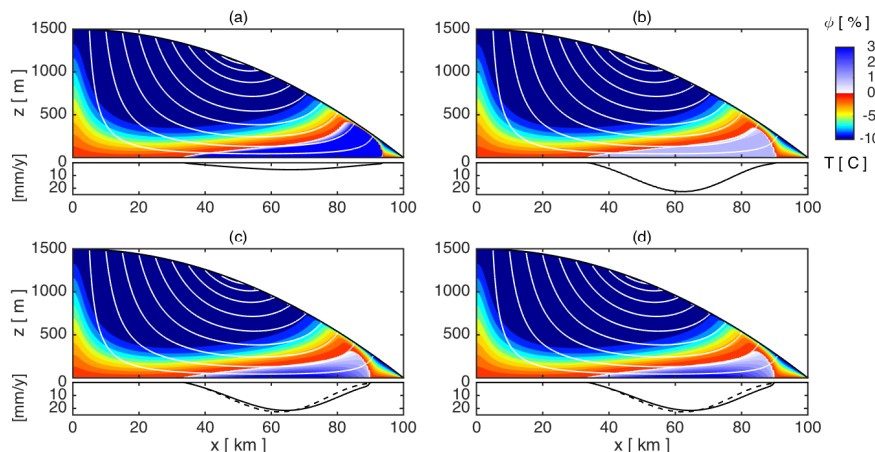

**Figure 7.** Ice cap solutions for (a,b) the standard enthalpy gradient model without and with drainage, (c) modified enthalpy gradient model, (d) compaction pressure model. Shading shows temperature and porosity. Solid white lines show ice streamlines. The lower panels show water flux from the ice at the bed; dashed lines are replicas of that from (b) for comparison. Parameter values are as in Table 1, together with $T_s = -10$ C, $T_m = 0$ C, and $p_e = 0$ at $z = 0$.

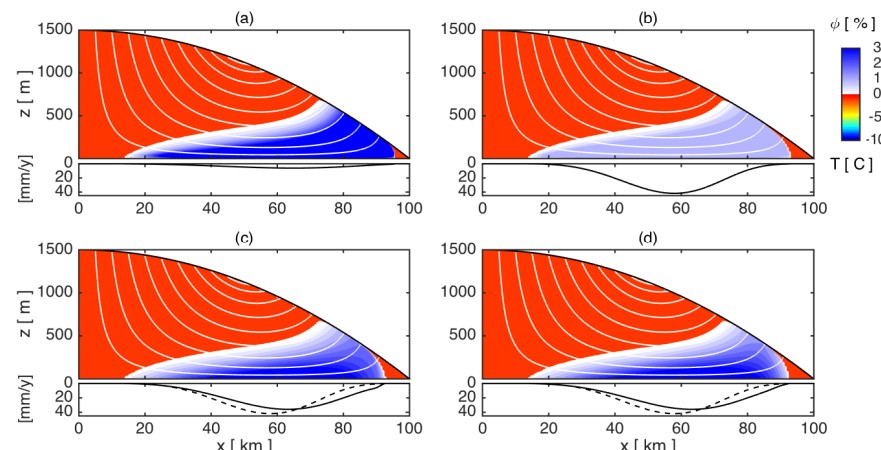

**Figure 8.** Ice cap solutions as in Figure 7 but with warmer surface temperature $T_s = -1$ C, showing an enlarged temperate region.



## 5   Discussion and Conclusion

### 5.1   Summary

We have presented methods for modelling the water content of temperate ice that make use of Darcy's law and thus allows water to drain under gravity. The models provide an alternative to commonly used enthalpy gradient methods, and we have shown that they can be solved in a similar way using an enthalpy-based formulation. The gravity-driven water fluxes are coupled to latent heat transport in temperate ice, so can result in changes to the location of cold-temperate interfaces from what is predicted by other models. The new models correctly handle jump conditions at both melting and freezing cold-temperate interfaces.

We envisage that the compaction pressure method could quite easily be implemented in existing ice-sheet models, requiring many fewer alterations than a full two-phase formulation. Alternatively, the 'modified' enthalpy gradient method has been shown to give very similar results in many cases. That model assumes that the pore pressure is equal to the ice pressure, and the buoyancy $(\rho_w - \rho)\mathbf{g}$ therefore drives water flow through the permeable ice. In practice there is little difference between this method and the introduction of the drainage function to the standard model; $\nabla \cdot [k_0 \phi^\alpha (\rho_w - \rho)\mathbf{g}/\eta_w]$ serves essentially the same role as $D(\phi)$. The chief advantage is being able to track the passage of water through the ice.

Although the compaction pressure model requires the addition of a new variable, the effective pressure, the computational cost of this new variable is relatively small. In fact, the compaction pressure model was often cheaper to solve than the modified enthalpy gradient model using our algorithm, the latter generally requiring smaller time steps to be taken. The advantage of including the effective pressure is that it allows for a connection with the subglacial drainage system, and for the possibility that the pore water pressure becomes close to hydrostatic (in which case gravity-driven drainage does not occur as rapidly as one might otherwise think).

### 5.2   Comments on the compaction pressure model

The compaction pressure model is essentially a stripped down version of the more detailed theory presented by Fowler (1984); there is a subtle difference in the boundary conditions applied at the cold-temperate interface, which would also become more complicated in our model if the density difference between ice and water were considered in full. The model has similarities with related literature on magma extraction from the mantle (McKenzie, 1984).

The permeability of the temperate ice evidently plays an important role in this theory. It is unlikely that a particular value for $k_0$ would be correct in all circumstances, since the crystal structure of the ice (particularly near the bed) may be highly variable, and the presence of impurities may change the surface energy and thus the microscopic arrangement of the pore space (Lliboutry, 1996). We have chosen a value largely on the basis of how much water it allows to drain, the observation being that



large porosities (a few %) are rare. It is nevertheless instructive to realise how the behaviour of the model differs depending on the magnitude of the permeability (Figure 3). When the permeability is small, the role of the effective pressure in driving water flow is generally confined to boundary layers near the edges of the temperate ice, and the flux is well approximated by just the first gravitational term in (14) (though see below). When the permeability is larger, the effective pressure gradient balances this first term, and this gives rise to nearly hydrostatic conditions.

The length scale over which the effective pressure gradients play a role is seen from (15) to be

$$\ell = \left( \frac{[\eta]k_0[\phi]^{\alpha-1}}{\eta_w} \right)^{1/2}, \tag{25}$$

where $[\eta]$ and $[\phi]$ are typical values of these variables. This length scale is referred to in the magma-dynamics literature as the 'compaction length'. For parameters in Table 1 and $[\eta] \sim 10^{13}$ Pa s, $[\phi] \sim 0.01$, it is around 10 m, and therefore small compared to the ice-sheet scale, though this depends quite sensitively on the permeability as discussed above.

The nature of the solutions in the case of a small compaction length is explored by Schoof and Hewitt (2016). One particular feature is relevant to the parallel slab situation in Figure 2(c); when gravity-driven drainage opposes the direction of ice advection, the effect of a lower boundary is not just confined to a boundary layer, but extends far into the temperate region. This is because the imposed permeability of the ice at the boundary is inconsistent with the gravitationally-driven downward water flux $k_0\phi^\alpha(\rho_w - \rho)g/\eta_w$, and large effective pressure gradients are required to enable the flow. The large effective pressures drive compaction and de-compaction of the ice, which results in a band of large porosity above the boundary as seen in Figure 2(c). This situation of upwards advection into a temperate region is unlikely to be common, if indeed it ever occurs; but one can imagine that such behaviour might occur in an overdeepening, where de-pressurisation results in freeze-on of subglacial water. The margins of ice stream are another place where this may be relevant.

### 5.3 Extensions

We end by commenting on straightforward extensions to this model. One possibility is to include the term $-\nabla \cdot \mathbf{j}$ in (4) as part of the Stokes flow solution, to account for the compression of the ice as a result of internal water drainage. This is relatively straightforward if a depth-integrated approximation of the Stokes equations is used, since in that case (4) is used only to determine the vertical velocity $w$, having solved the momentum equations for the horizontal components $u$ and $v$. In this case,

$$w = w_b - \int_{Z_b}^{z} \left( \frac{\partial u}{\partial x} + \frac{\partial v}{\partial y} + \hat{\nabla} \cdot \hat{\mathbf{j}} \right) \mathrm{d}z, \tag{26}$$

where $w_b$ is the vertical velocity at $z = Z_b$ (due to melting). The inclusion of the drainage term here provides an additional coupling, along with porosity dependence of the flow parameter $A$, between



water drainage and ice dynamics. However, this is still only an approximate method to account for
the ice compression, since the non-zero divergence of ice velocity means that the strain rate tensor
$\dot{\varepsilon}_{ij}$ is no longer traceless, and so a modification to the flow law (3) is strictly also required (see
Schoof and Hewitt (2016)).

A similar amendment that would be straightforward in a coupled model is to include the deviatoric
stress components in Darcy's law (12). These amount to an additional pressure gradient $\nabla p_r$ in (14),
which effectively modifies the direction and magnitude of gravity (Schoof and Hewitt, 2016). The
corrections are generally expected to be small, but there could be locations where water flow is
driven in quite a different direction to the vertical.

The fact that deviatoric stresses in the ice could lead to differential compaction of different orientations of veins (Nye and Mae, 1972), and thus an anisotropic permeability, would add greater
complexity. The role of impurities (Lliboutry, 1971), and of thermodynamic pre-melting (Wettlaufer
and Worster, 2006), also require further consideration. Finally, we should note that the applicability
of simple porous media flow to polythermal ice is not guaranteed, and additional physics such as
brittle fracturing may play a role. If liquid water transport occurs through macroscopic veins, cracks,
or crevasses, rather than through inter-granular veins, one can imagine that on a large scale this
might still be treated as a porous flow, but with a different interpretation of permeability. There may
also be the possibility of flow-driven enlargement of cracks similar to the channelisation that occurs
subglacially.

*Acknowledgements.* IJH is supported by a Marie Curie FP7 Career Integration Grant within the 7th European
Union Framework Programme. CS acknowledges the support of NSERC grants 357193-13 and 446042-13 and
a Killam Faculty Research Fellowship at UBC.





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
