# Peer review of "Models for polythermal ice sheets and glaciers"

_The Cryosphere, 2016_

## Referee Comment (RC1) · Anonymous Referee #1 · 1 Dec 2016

**Review of *"Models for polythermal ice sheets and glaciers"* by Hewitt and Schoof**

This paper is a simplified presentation of a mathematical model already described in *Journal of Fluid Dynamics* by the same authors [*Schoof and Hewitt*, 2016]. It clearly makes the model more accessible to the cryospheric science community, making easier the incorporation of this new approach in computational glaciology. The model itself is a very nice improvement of the mathematical representation of polythermal glaciers by improving the modeling of the water transport in the temperate ice. The strength of the model is also to stay simple enough to be easily implemented on current thermo-mechanical ice-flow models.

The approach uses the enthalpy formulation [*Aschwanden et al.*, 2012] that allows the use of an unique variable for both temperate and cold part of the ice. This avoids explicitly representing the CTS as a boundary condition and having two independent systems to solve in cold and temperate ice. Only the enthalpy gradient method has been used so far for glaciers. It assumes a diffusive water flux proportional to the enthalpy gradient which can lead to unrealistically high water content. The method requires therefore a drainage parametrization to cap the water content to an imposed value. In this study, the authors introduce a water transport according to Darcy Law and driven by gravity and water pressure gradient using the assumption of viscous compaction rate to compute pore pressure. The two main advantages are the water drainage is physically computed and the pressure gradient allows connecting water content in temperate ice to a subglacial hydrology model that would provide the adequate boundary condition at the bed.

The authors explore three different approaches through very pedagogic examples that clearly help to understand the difference between these approaches. They also conclude that just adding gravity driven water flow in the current enthalpy gradient method leads to very satisficing results compared to the full gravity and pressure gradient method (which need to solve a supplementary variable).

The paper is well written and structured. It is complementary to *Schoof and Hewitt* [2016] making easier the access of the model to the community. I think the manuscript deserves publication in The Cryosphere with few minor revisions.

**General comments:**

- Because mathematical and numerical aspect of the model have already been treated in *Schoof and Hewitt* [2016] and the aim of this paper is to communicate to glaciologist community, I think it would be better if this paper tries to link more the modeling results to observation of the "real" world. I know that very few observations of water flux through temperate ice have been done which gives lot of freedom to modeler… At least the authors should discuss in the introduction why water flow in temperate ice could be modeled as a porous flow by referring more to literature on this subject. Also it would be needed in the introduction to describe the difference between your approach and the existing ones. For exemple, *Fowler* [1984] also used a darcy law… I know this have been done more carefully in *Schoof and Hewitt* [2016] but I think it should appear also in this paper.

- The natural boundary condition at the base of a glacier for energy conservation equation is generally a heat flux coming from the ground. You should show an example using this type of boundary condition at least for the ice cap setting (probably instead of figure 8 which is not very useful).

- Why using *v = (k/ρc)/100* while *Aschwanden et al.* [2012] recommend *v = (k/ρc)/10* and most of people are using this regularization then. It would be better to compare enthalpy gradient method with a more "standard" *v* parameter.

**Specific comments**

*Section 2.3:* I would present first the standard enthalpy gradient model, then the modified enthalpy gradient model and finally the compaction pressure model. It would avoid referring to section 2.3.2 and 2.3.3 in section 2.3.1.

*Line 124:* Discuss more about the value of $k_0$. Do you think large value of $k_0$ would be an adequate way to model water transport through macroscopic veins and crack? Actually, may be add this in section 5.2 and not here.

*Line 174 -177*: Any idea about $p_e'(\phi)$ ? May be develop a bit more about a model using $v(\phi)$ rather than constant *v.*

*Line 187-189:* I guess it is because $\phi=0$ in the cold part?

*Line 274:*  porosity : infinite porosity ?

*Figure 3:* In the legend, you say for large permeability dPe/dz ≈ $(\rho_w-\rho)g$ from (14). I don't understand how you get this from (14).

*Line 304-308:* Could you explain more why pressure become hydrostatic in the model for large permeability. Also it is not clear to me why drainage is controlled by effective pressure

in this case? Lot of the drainage still occurs via gravity… Do you mean porosity is controlled by effective pressure?

Line 351:

Line 410: Still not clear to me why effective pressure gradient balance gravitational term for large permeability?

Line 422: permeability: do you mean porosity here ?

Line 424: effective pressure gradient

Line 428: What do you mean by the margins of ice stream are another place where this may be relevant. Be more precise.

Line 451-457: Add a ref like [*Fountain and Walder*, 1998]

**Reference:**

Aschwanden, A., E. Bueler, C. Khroulev, and H. Blatter (2012), An enthalpy formulation for glaciers and ice sheets, *J. Glaciol.*, *58*(209), 441–457, doi:10.3189/2012JoG11J088.

Fountain, A. G., and J. S. Walder (1998), Water flow through temperate glaciers, *Rev. Geophys.*, *36*(3), 299–328, doi:10.1029/97RG03579.

Fowler, A. C. (1984), On the transport of moisture in polythermal glaciers, *Geophys. Astrophys. Fluid Dyn.*, *28*(2), 99–140, doi:10.1080/03091928408222846.

Schoof, C., and I. J. Hewitt (2016), A model for polythermal ice incorporating gravity-driven moisture transport, *J. Fluid Mech.*, *797*, 504–535, doi:10.1017/jfm.2016.251.

---

## Referee Comment (RC2) · A. Aschwanden (Referee) · 19 Dec 2016

**Review of "Models for polythermal ice sheets and glaciers" by Hewitt & Schoof**

**December 18, 2016**

The manuscript under review is the long awaited extension and refinement of Aschwanden et al. (2012). I've never been satisfied with using a Fick-type diffusion for the water flux and I'm very delighted that Hewitt and Schoof present a more physical Darcy-type model. Hewitt and Schoof's work draws upon earlier work by Fowler (1984) but goes beyond because it describes two Darcy models that are suitable for implementation in existing models, making the manuscript an even more useful contribution. I agree with the authors that it is not obvious how much the new methods will improve realistic ice sheet models simulations, especially with the sparse observations available for validation in mind. I anticipate we will implement the modified enthalpy gradient method some time next year. One thing that is not quite clear to me is if, in a coupled model, it will be necessary to implement a cap on the porosity to avoid reaching nonphysical values in areas with high strain heating.

The paper is well written and easy to follow; it is basically publishable as-is, I only have a few trivial comments below.

Cheers, Andy Aschwanden

**Technical Comments**

**L18** "polythermal ice masses"

**Eq. 14 and 15** Maybe use `\left(...\right)` to make the outer parentheses a bit bigger?

**Figures** The figures are of high quality and, in general, easy to read. Adding a legend to Figs 2,3 would further increase readability (while the information which line is which is in the caption, a legend makes it more straightforward)

**References**

Aschwanden, A., E. Bueler, C. Khroulev, and H. Blatter (2012). An enthalpy formulation for glaciers and ice sheets. *J. Glaciol.*, **58**(209), 441–457. doi: 10.3189/2012JoG11J088.

Fowler, A. C. (1984). On the transport of moisture in polythermal glaciers. *Geophys. Astro. Fluid*, **28**(2), 99–140. doi: 10.1080/03091928408222846. URL http://www.informaworld.com/openurl?genre=article&doi=10.1080/03091928408222846&magic=crossref%7C%7CD404A21C5BB053405B1A640AFFD44AE3.

---

## Author Comment (AC1) · 21 Dec 2016

**Response to reviewer comments**

Since most of the referees' requests for changes are relatively minor, we have responded to their comments below (in red) by describing how the manuscript has been revised, in the hope we may submit a revised version.

**Referee 1**

This paper is a simplified presentation of a mathematical model already described in Journal of Fluid Dynamics by the same authors [Schoof and Hewitt, 2016]. It clearly makes the model more accessible to the cryospheric science community, making easier the incorporation of this new approach in computational glaciology. The model itself is a very nice improvement of the mathematical representation of polythermal glaciers by improving the modeling of the water transport in the temperate ice. The strength of the model is also to stay simple enough to be easily implemented on current thermo-mechanical ice-flow models.

The approach uses the enthalpy formulation [Aschwanden et al., 2012] that allows the use of an unique variable for both temperate and cold part of the ice. This avoids explicitly representing the CTS as a boundary condition and having two independent systems to solve in cold and temperate ice. Only the enthalpy gradient method has been used so far for glaciers. It assumes a diffusive water flux proportional to the enthalpy gradient which can lead to unrealistically high water content. The method requires therefore a drainage parametrization to cap the water content to an imposed value. In this study, the authors introduce a water transport according to Darcy Law and driven by gravity and water pressure gradient using the assumption of viscous compaction rate to compute pore pressure. The two main advantages are the water drainage is physically computed and the pressure gradient allows connecting water content in temperate ice to a subglacial hydrology model that would provide the adequate boundary condition at the bed. The authors explore three different approaches through very pedagogic examples that clearly help to understand the difference between these approaches. They also conclude that just adding gravity driven water flow in the current enthalpy gradient method leads to very satisficing results compared to the full gravity and pressure gradient method (which need to solve a supplementary variable).

The paper is well written and structured. It is complementary to Schoof and Hewitt [2016] making easier the access of the model to the community. I think the manuscript deserves publication in The Cryosphere with few minor revisions. We thank the referee for their careful reading of the manuscript and the helpful suggestions below.

General comments:

- Because mathematical and numerical aspect of the model have already been treated in Schoof and Hewitt [2016] and the aim of this paper is to communicate to glaciologist community, I think it would be better if this paper tries to link more the modeling results to observation of the "real" world. I know that very few observations of water flux through temperate ice have been done which gives lot of freedom to modeler? At least the authors should discuss in the introduction why water flow in temperate ice could be modeled as a porous flow by referring more to literature on this subject. Also it would be needed in the introduction to describe the difference between your approach and the existing ones. For exemple, Fowler [1984] also used a darcy law? I know this have been done more carefully in Schoof and Hewitt [2016] but I think it should appear also in this paper.

We have added more discussion to the introduction (a new third paragraph), and added more specific referencing to the existing approaches.

- The natural boundary condition at the base of a glacier for energy conservation equation is generally a heat flux coming from the ground. You should show an example using this type of boundary condition at least for the ice cap setting (probably instead of figure 8 which is not very useful).

We don't fully agree with this comment. The natural boundary condition is a heat flux if the ice is below the melting temperature, but if the ice reaches the melting temperature the boundary condition is fixed temperature (at melting point), while the heat flux goes into calculating the basal melt rate (or freezing rate) as part of a Stefan condition. The geothermal heating is therefore more important for calculating the basal melt rate than for the temperature above the bed. We didn't want to have a drawn-out discussion of this in the paper because it gets a bit fiddly - in particular one needs to add a model of the basal water layer (because you need to know how much water is around at each location to know whether basal freezing can occur - which sustains the ice at the melting point when one might otherwise have thought that the temperature should be below melting point).

In the figure, we assume that there is sufficient heat flux from the ground to ensure that the base of the ice is at the melting point. We have added further discussion (in the second paragraph of section 4.3) that one can also have conditions of fixed heat flux when ice is below the melting point. I think this issue is best treated when the model is coupled with a basal hydrology model, and preferably with a solution for the temperature in the upper layers of the substrate too. We have therefore left figure 8 as it is.

- Why using $\nu = (k/\rho c)/100$ while Aschwanden et al. [2012] recommend $\nu = (k/\rho c)/10$ and most of people are using this regularization then. It would be better to compare enthalpy gradient method with a more "standard" $\nu$ parameter.

Thanks for pointing this out. We used the 1/100th value as illustration, and were unaware that 1/10th was conventional. We have re-done the calculations with $\nu = (k/\rho c)/10$ and find that the solutions for the enthalpy gradient method are similar but with considerably larger boundary layer features (as would be expected). Since we want to emphasise the fact that the limit nu goes to zero approaches the no-water-transport solution we have retained the figures with $(k/\rho c)/100$, but add the comment that the boundary layers are larger with $\nu = (k/\rho c)/10$. The relevant figures with larger $\nu$ are included at the end of this response for comparison.

Specific comments

Section 2.3: I would present first the standard enthalpy gradient model, then the modified enthalpy gradient model and finally the compaction pressure model. It would avoid referring to section 2.3.2 and 2.3.3 in section 2.3.1.

We prefer not to do this re-ordering, because the natural place to introduce the permeability and the buoyancy-driven water flow that enter in the modified enthalpy graidnet method is in Darcy's law, which is only used in its complete form in the compaction pressure model. Thus we prefer to present that model first, so as to progressively introduce the ingredients.

Line 124: Discuss more about the value of $k_0$. Do you think large value of $k_0$ would be an adequate way to model water transport through macroscopic veins and crack? Actually, may be add this in section 5.2 and not here.

We have added to section 5.3: "If liquid water transport occurs through macroscopic veins, cracks, or crevasses, rather than through inter-granular veins, one can imagine that on a large scale this might still be treated as a porous flow, but with a different interpretation of permeability. In the model, this would correspond to taking a large value of $k_0$, in which case the compaction pressure model predicts efficient gravity-driven drainage and a hydrostatic 'pore'-pressure (consistent with the idea of an englacial water table)."

Line 174 -177: Any idea about $p'_e(\phi)$? May be develop a bit more about a model using $\nu(\phi)$ rather than constant $\nu$.

We have added a sentence at the end of section 2 to discuss this. Given the lack of empirical knowledge about $p_e(\phi)$ there does not seem justification to consider $\nu(\phi)$ explicitly (if $\nu$ were to increase dramatically at small $\phi$, which is possible, then the behaivour of the solutions could change somewhat, but provided nu stays 'small' its precise value does not significantly affect the solutions).

Line 187-189: I guess it is because $\phi = 0$ in the cold part?

Yes - clarified.

Line 274: finite porosity : infinite porosity ?

Yes - changed to 'unbounded' (though that is only for the no-water-motion problem, which is unrealistic; it is bounded in each of the other models).

Figure 3: In the legend, you say for large permeability $dP_e/dz \approx (\rho_w - \rho)g$ from (14). I don't understand how you get this from (14).

This is because if the permeability is large then the right hand side of (14) is large relative to the left hand side (the flux, whose size is limited by how much water is being produced), UNLESS the term in brackets is close to zero; which corresponds to approximate hydrostatic balance. That is $dp_e/dz = (\rho_w - \rho)g$ corresponds to $dp/dz = \rho_w g$. The 'pore space' acts rather like a moulin in this limit; that is, it provides so little resistance to flow through it that the pressure is approximatley hydrostatic. It is not exactly hydrostatic, because a small deviation is required to drive the flow downwards through it, but the flow rate would be vastly overestimated if the potential gradient $(\rho_w - \rho)g$ were used to estimate it.

It is true that gravity is still ultimately responsible for the drainage. The point here was that there is not a balance between the left hand side and the density terms in (14), as in the case of small permeability; the small deviations from hydrostatic balance that drive the water flow are controlled by the effective pressure in (15). But we realise that the comments on line 308 may be a misleading way of phrasing these dynamics. Ultimately in these steady-state solutions the rate of drainage is controlled by the rate of production (effective pressure and porosity adjust so as to squeeze the water out at the required rate). So we have

removed the sentence and simply commented that the pressure becomes approximately hydrostatic in the large permeability limit.

Line 304-308: Could you explain more why pressure become hydrostatic in the model for large permeability. Also it is not clear to me why drainage is controlled by effective pressure in this case? Lot of the drainage still occurs via gravity? Do you mean porosity is controlled by effective pressure?

See above

Line 351: there

OK

Line 410: Still not clear to me why effective pressure gradient balance gravitational term for large permeability?

See above

Line 422: permeability: do you mean porosity here ?

They are equivalent, but yes, perhaps that is more consistent.

Line 424: effective pressure gradient

No; we meant effective pressure itself, which is responsible for squeezing or dilating the pore space according to equation (15).

Line 428: What do you mean by the margins of ice stream are another place where this may be relevant. Be more precise.

Sentence expanded to explain this: "The margins of ice stream are another place where this may be relevant, if lateral advection brings cold ice underneath ice that has been heated in the shear margin."

Line 451-457: Add a ref like [Fountain and Walder, 1998]

OK

**Referee 2**

The manuscript under review is the long awaited extension and refinement of Aschwanden et al. (2012). I've never been satisfied with using a Fick-type diffusion for the water flux and I'm very delighted that Hewitt and Schoof present a more physical Darcy-type model. Hewitt and Schoof?s work draws upon earlier work by Fowler (1984) but goes beyond because it describes two Darcy models that are suitable for implementation in existing models, making the manuscript an even more useful contribution. I agree with the authors that it is not obvious how much the new methods will improve realistic ice sheet models simulations, especially with the sparse observations available for validation in mind. I anticipate we will implement the modified enthalpy gradient method some time next year. One thing that is not quite clear to me is if, in a coupled model, it will be necessary to implement a cap on the porosity to avoid reaching nonphysical values in areas with high strain heating. The paper is well written and easy to follow; it is basically publishable as-is, I only have a few trivial comments below.

Cheers, Andy Aschwanden

Thanks for the comments. In principle the ability for water to drain under gravity in proportion to an increasing power of the porosity in the modified enthalpy gradient method should prevent large porosities, even in areas with high strain heating. However it is possible that if a region of impermeable (or nearly impermeable) ice is beneath such areas, then a build up of porosity is possible and without a full two-phase treatment of the continuity equation (which would allow macroscopic liquid water bodies to form) this might cause problems. Thus I would probably recommend an additional cap on the porosity for practical purposes.

Technical Comments

L18 "polythermal ice masses"

OK

Eq. 14 and 15 Maybe use (...) to make the outer parentheses a bit bigger?

OK

Figures The figures are of high quality and, in general, easy to read. Adding a legend to Figs 2,3 would further increase readability (while the information which line is which is in the caption, a legend makes it more straightforward)

OK

[Figure]

**Figure 1.** Figure 2 from the paper but with larger $\nu = (k/\rho c)/10$.

[Figure]

**Figure 2.** Figure 5 from the paper but with larger $\nu = (k/\rho c)/10$.

[Figure]

**Figure 3.** Figure 7 from the paper but with larger $\nu = (k/\rho c)/10$.